# Differences and Associations of NLRP3 Inflammasome Levels with Interleukins 1α, 1β, 33 and 37 in Adults with Prediabetes and Type 2 Diabetes Mellitus

**DOI:** 10.3390/biomedicines11051315

**Published:** 2023-04-28

**Authors:** Hend Alfadul, Shaun Sabico, Mohammed G. A. Ansari, Abdullah M. Alnaami, Osama E. Amer, Syed D. Hussain, Kaiser Wani, Malak N. K. Khattak, Mario Clerici, Nasser M. Al-Daghri

**Affiliations:** 1Chair for Biomarkers of Chronic Diseases, Biochemistry Department, College of Science, King Saud University, Riyadh 11451, Saudi Arabia; 2Biochemistry Department, College of Science, King Saud University, Riyadh 13579, Saudi Arabia; 3Department of Medical-Surgery Physiopathology and Transplantation, University of Milan, 20122 Milan, Italy; 4Don C. Gnocchi Foundation ONLUS, IRCCS, 20122 Milan, Italy

**Keywords:** T2DM, inflammation, interleukin

## Abstract

Inflammasome activation of the nucleotide-binding domain, leucine-rich–containing family, and pyrin domain–containing-3 (NLRP3) has been observed to be involved in the pathogenesis of numerous inflammatory diseases, including prediabetes (PD) and type 2 diabetes mellitus (T2DM). Varying levels of glycemia can trigger inflammasome activation; yet, limited studies have reported the associations between NLRP3 levels or other circulating interleukins (ILs) and glycemic status. This study investigated the differences and associations between serum levels of NLRP3 and IL-1α, IL-1β, IL-33 and IL-37 in Arab adults with PD and T2DM. A total of 407 Saudi adults (151 males and 256 females) (mean age = 41.4 ± 9.1 years and mean BMI = 30.7 ± 6.4 kg/m^2^) were included. Overnight-fasting serum samples were collected. The participants were stratified according to T2DM status. Serum levels of NLRP3 and ILs of interest were assessed using commercially available assays. In all participants, age- and BMI-adjusted circulating levels of IL-37 were significantly higher in the T2DM group (*p* = 0.02) than in healthy controls (HC) and the PD group. A general linear model analysis revealed that NLRP3 levels were significantly influenced by T2DM status; age; and ILs 18, 1α and 33 (*p*-values 0.03, 0.04, 0.005, 0.004 and 0.007, respectively). IL-1α and triglycerides significantly predicted NLRP3 levels by as much as 46% of the variance perceived (*p* < 0.01). In conclusion, T2DM status significantly influenced NLRP3 expression and other IL levels in varying degrees. Whether these altered levels of inflammasome markers can be favorably reversed through lifestyle interventions needs to be investigated prospectively in the same population.

## 1. Introduction

Type 2 diabetes mellitus (T2DM) is a global epidemic and one of the top ten causes of mortality around the world [1]. In the Middle East, particularly in the Kingdom of Saudi Arabia (KSA), one out of every four adults has diabetes, mainly T2DM. By 2031, the incidence of T2DM in KSA is expected to double [2]. T2DM is a multi-factorial non-communicable disorder that is notorious for causing insulin resistance and long-lasting sub-clinical inflammation with elevated systemic levels of adipokines, tumor necrosis factor (TNF) and interleukins (IL) [3]. A strong driver of T2DM is an abnormal increase in adiposity, also known as obesity. Altered adipocytes secondary to obesity have been observed to overexpress hormones and inflammatory cytokines, causing obese individuals to be highly susceptible to long-term chronic inflammation [4]. Thus, increased adipocytes negatively amplify the inflammatory response, leading to insulin resistance and, ultimately, the development of T2DM [2]. Since obesity is correlated with the stimulation of the innate immune system, the analysis of various inflammatory markers is highly significant in managing and treating T2DM [4]. Other significant risk factors of T2DM include age and sex. Epidemiological data suggest that the prevalence of metabolic disorders, including diabetes, around the world—except in the Middle East and North Africa—is higher in middle-aged males than females, despite females being more predisposed to obesity [5]. This pattern reveals sexual dimorphism in the pattern and progression of the disease.

A strong effector on the host’s defense mechanism is the innate immune system, which is triggered by pattern-recognition receptors (PRRs). PRRs stimulate inflammation by identifying pathogen-(PAMPs) and danger-associated molecular patterns (DAMPs), both of which are capable of initiating an immune response [6]. Martinon and colleagues discovered inflammasomes back in 2002, and found that they play a major role in innate immunity [7]. Cytosolic inflammasomes are cytosolic innate immune PRRs present in various cell types that sense DAMPs and PAMPs responsible for triggering inflammation [6,7,8].

Among the roster of inflammasomes, NLRP3, or nucleotide-binding oligomerization domain, leucine-rich repeat and pyrin domain-containing protein-3, is the most commonly studied inflammasome complex [3]. Since it can be stimulated in response to various activators, the NLRP3 inflammasome is regarded as the most aggressive inflammasome [9]. The NLRP3 inflammasome complex consists of three different proteins: (1) NLRP3, which is the sensing PRR protein; (2) ASC, or apoptosis-associated speck-like protein, containing a caspase recruitment domain which is the connecting protein; and (3) peo-caspase-1, which is the acting protein [3]. In addition to its own release, caspase-1 is the enzyme that regulates the release of different interleukins (IL) which do not have release signals. The caspase-1 secretome consists of the well-analyzed IL-18 and IL-1β, plus the proteins that are not immediate protease substrates, such as IL-1α [10]. Upon receiving an activation signal, NLRP3 inflammasome complex proteins oligomerize and polymerize, forming an active NLRP3 inflammasome complex which serves as the stage for IL-18 and IL-1β synthesis [3]. The caspase-1 secretome proteins are involved primarily in tissue repair, cell protection and inflammation [10].

Hyperactivation of the NLRP3 complex has been associated with the pathogenesis of many autoimmune and inflammatory disorders, including obesity, cardiovascular disease and T2DM [11,12]. Hyperglycemia, a diabetes-related endogenous DAMP [13], can provide a priming signal in NLRP3 inflammasome activation, increasing the transcription of the IL-1β gene and, ultimately, the secretion of pro-ILs. Hyperglycemia can also increase the production of reactive oxygen species (ROS), providing the second signal, called the activation signal, as well as initiating NLRP3 complex assembly and active IL-18 and IL-1β production [14]. In our recent systematic review and meta-analysis [3], serum IL-1β levels were mainly found to be increased in the T2DM group, and were positively correlated with hemoglobin A1 concentration (HbA1c) and fasting blood glucose. Despite those findings, limited data are available that could characterize IL-1β as a strong biomarker for T2DM, despite being a promising biomarker for insulin-related disorders.

Despite universal agreement from different studies showing the vital role of controlling the blood glucose level in disease outcomes, the means by which it aggravates disease development has not yet been completely clarified [15]. One hypothesis is that fluctuations in blood glucose levels increase ROS levels and the production of inflammatory cytokines, but this is not yet fully understood [16]. Thus, in this cross-sectional study, we aimed to investigate, for the first time, the serum levels of NLRP3 with pro-inflammatory and anti-inflammatory ILs (1A, 1β, 33 and 37) in Saudi adults with different levels of glycemia, prediabetes (PD) and T2DM, and to compare them to healthy controls (HC). Since T2DM is associated with chronic subclinical inflammation, it is also of interest to note the associations between circulating levels of NLRP3 and the established ILs.

## 2. Materials and Methods

### 2.1. Study Design and Participants

In this cross-sectional study, clinical data obtained from 407 (151 males and 256 females) Saudi adults (age = 41.4 ± 9.1 years and BMI = 30.7 ± 6.4 kg/m^2^) were included. Information on the participants as taken from the master database of the Chair for Biomarkers of Chronic Diseases (CBCD), College of Science in King Saud University (KSU), Riyadh, KSA, a collaborative registry with the Ministry of Health that includes clinical information and blood samples of more than 10,000 Saudis aged 1–99 years who have been invited from various primary care centers (PCCs) in Riyadh, KSA [17,18]. The exclusion criteria that were taken into consideration included pre-existing cardiovascular disease, inflammatory systemic disease, eating disorders, celiac disease, inflammatory bowel disease, systemic glucocorticoid use (one month of cumulative use during the year prior to inclusion), non-consenting individuals, those who were pregnant and anyone with multi-organ complications. A written informed consent was obtained from all participants prior to inclusion in the database (project # E-20-5127). Ethical approval was obtained from the College of Science Research Center, KSU, Riyadh, KSA (Ref 21/0412/IRB). A general questionnaire on demographic information, general health status and past medical history was completed by all participants.

### 2.2. Sample Collection and Anthropometrics

Blood and serum samples from the selected participants were taken from the Biobank database of CBCD in KSU. In brief, these consisted of overnight fasting whole-blood and serum samples collected from all participants by nurses and technicians trained in their respective PCCs. To obtain serum samples from whole blood, blood was allowed to clot undisturbed in red-top tubes at (25 °C) for between a minimum of half an hour and a maximum of one hour, then centrifuged at 1000 to 1500× *g* for 10 min, using a refrigerated centrifuge (4 °C) to remove the clot. After centrifugation, the obtained serum was then directly aliquoted, clearly labeled and delivered to CBCD in KSU for immediate storage (−80 °C) until further analysis. Anthropometrics, which included weight (kg), height (cm), waist and hip circumferences (cm) and blood pressure (mmHg) assessments, were taken by trained nurses. Body mass index (BMI) (kg/m^2^) and waist–hip ratio (WHR) were obtained using the following equations: weight in kilograms divided by the square of height in meters for BMI, and quotient between waist and hip circumferences for waist–hip ratio (WHR). The T2DM criteria were based on those of the American Diabetes Association (ADA): a fasting serum glucose (FSG) less than 5.55 mmol/L was considered normal, between 5.55 to 6.9 mmol/L was considered PD and higher than 6.9 mmol/L was considered T2DM [19]. Participants were stratified based on their fasting glucose levels. Group 1 (normoglycemic healthy controls, HC) included 198 participants (39.3 ± 8.9 years), group 2 (prediabetes, PD) included 88 participants (42.3 ± 8.8 years) and group 3 (T2DM) included 121 participants (44.0 ± 8.7 years).

### 2.3. Lipid Profile and Biochemical Estimations

Serum samples from the selected participants were thawed prior to biochemical assessments. Serum triglycerides, total cholesterol, high-density lipoprotein (HDL) and cholesterol and glucose levels were analyzed routinely using commercially purchased kits (catalog nos. 981379, 981812, 981823 and 981301, respectively) and an automated biochemical analyzer (Konelab 20 Thermo Fischer, Espoo, Finland) [8]. HbA1c was analyzed directly using ion exchange high-performance liquid chromatography (HPLC) with commercially available reagents (D-10 Hemoglobin A1c Program #12000949) in a fully automated system (D-10 Hemoglobin Testing System #12010405, Bio-Rad, Hercules, CA, USA). Fasting serum insulin levels were analyzed using a Luminex multiplex system (Luminex Corp, Austin, TX, USA) and a commercially available kit, catalog no. HINS-MAG. The intra-assay % CV was <10 and the inter-assay % CV was <15, according to the manufacturer’s protocol.

### 2.4. Serum NLRP3 and ILs (1α, 1β, 33 and 37) Estimations

Circulating levels of NLRP3 was estimated using enzyme-linked immunosorbent assays (ELISA), catalog number CSB-E15885h, Cusabio, Houston, TX, USA. According to the manufacturer’s protocol, the minimum detectable dose for this assay was <0.039 ng/mL of human NLRP3 with a CV% of <8% and <10% for intra- and inter-assay precision, as previously described [8]. Circulating levels of IL-1β, IL-18 and IL-37 were assessed using the Flex MAP-3D System (Luminex Corporation, Austin, TX, USA), which utilized human cytokine magnetic bead panels (IL-18 and IL-1β: HCYTA-60K human cytokine, chemokine growth factor panel A—immunology multiplex assay) (IL-37: HCYP4MAG-64K human cytokine chemokine magnetic bead panel IV). The intra- and inter-assay % CVs for IL-18, IL-1β and IL- 37 were <15 and <20 and <10 and <15, respectively. Circulating levels of IL-1α and IL-33 were estimated using ELISA assays (Bio Vendor, R & D systems, Brno, Czech Republic) Cat No. RAF045R and RAF064R, respectively. The intra-assay % CVs for both were <5.4% and 4.7%, respectively, while the inter-assay % CVs for both were <10% and 6.9%, respectively. The controls and standards implemented in all the biochemical assays were regularly reviewed by the quality assurance team of CBCD in KSU.

### 2.5. Statistical Analysis

The results were analyzed using SPSS (version 22, Chicago, IL, USA). Continuous variables are presented as means ± standard deviation (SD) for normal data, and median (25th and 75th) percentiles were used for non-normal variables. Categorical variables are shown as percentages (%). The Kolmogorov–Smirnov test was used to test for normality. Non-normal variables were log-transformed prior to analysis. One-way analysis of variance (ANOVA) and the Kruskal–Wallis H test were performed to compare mean and median differences in normal and non-normal variables between HC, PD and T2DM participants, respectively. Bonferroni’s post hoc multiple comparisons were performed between groups, and univariate analyses adjusted for age and BMI. Correlation analyses with NLRP3 and variables of interest were conducted using Pearson’s and Spearman analysis methods. Stepwise regression analysis, as well as a general linear model, were performed using parameters including NLRP3, ILs (18, 1-α, 1β, 33 and 37), age, BMI, waist–hip ratio, systolic blood pressure, diastolic blood pressure, glucose, HbA1c, insulin, total cholesterol, HDL cholesterol and triglycerides. NLRP3 and all ILs were included as dependent variables, and were also tested against each other in the analysis. *p* < 0.05 was considered significant.

## 3. Results

The clinical characteristics of all 407 participants are summarized in Table 1. The control group (HC) included 198 participants, the PD group included 88 participants and the T2DM group included 121 participants. In the T2DM group, systolic blood pressure (127.5 mmHg ± 16.2) and insulin (19.4 uU/mL (15.9–19.8)) were significantly higher (*p* = 0.001) than in the HC group (119.2 mmHg ± 12.0, and 12.7 uU/mL (6.3–18.9)) or in the PD group (122.4 mmHg ± 12.7; *p* < 0.001). In the PD and T2DM groups, HDL cholesterol levels (1.1 mmol/L ± 0.3 and 1.2 mmol/L ± 0.4, respectively) were significantly lower (*p* < 0.001) than the HC levels (1.3 mmol/L ± 0.4). In the PD and T2DM groups, triglyceride levels (1.7 mmol/L (1.2–2.3) and 1.7 mmol/L (1.3–2.2), respectively) levels were significantly higher (*p* < 0.001) than the in HC (1.2 mmol/L (0.9–1.7)). There were no significant differences in diastolic blood pressure or total cholesterol between any of the groups. *p*-values were adjusted for age and BMI.

In Figure 1A, IL-37 levels of all the participants in the T2DM group (6.01 (3.87–6.21)) were significantly higher than in the HC and PD groups (2.91 (2.2–3.8), 2.68 (2.1–7.6); *p* = 0.016), respectively. Figure 1B shows that IL-37 levels in the male T2DM group (6.14 (5.9–6.3)) were significantly higher than the HC and PD group levels (2.9 (2.11–3.01), 2.08 (2.0–2.7; *p* = 0.001), respectively. In Figure 1C, IL-37 levels in the female PD group (5.10 (2.12–10.9)) were significantly higher than in the HC group (2.92 (2.2–4.1); *p* = 0.03). Lastly, in Figure 1D, it is shown that NLRP3 levels in the female PD group (0.15 (0.11–0.32)) were significantly higher than those in the HC group (0.12 (0.07–0.18); *p* = 0.006).

In Figure 2A, there was a significant and positive association between log NLRP3 and log IL-33 (R = 0.38, *p* = 0.001). In Figure 2B, there was a significant positive association between log IL-37 and log NLRP3 (R = 0.144, *p* = 0.03).

Table 2 shows the stepwise regression analysis for NLRP3 and ILs (1-α, 1β, 33 and 37) (dependent variables, respectively) with age, BMI, WHR, blood pressure, glucose, HbA1c, insulin, total cholesterol, HDL cholesterol and triglycerides as independent parameters. IL-1α and triglycerides significantly predicted NLRP3 levels by as much as 46% of the variance perceived (*p* < 0.01), while triglycerides and NLRP3 significantly predicted IL-1α levels by as much as 43% of the variance (*p* < 0.01), and diastolic blood pressure significantly predicted IL-33 levels by 11% of the variance (*p* < 0.01). No significant predictors were obtained for ILs 1β and 37.

The results of the GLM analysis are presented in Table 3. For NLRP3, the levels were significantly influenced by T2DM status, age as well as ILs 18, 1α and 33 (*p*-values of 0.03, 0.04, 0.005, 0.004 and 0.007, respectively). IL-1α levels were significantly influenced by T2DM, IL-33 and NLRP3 (*p*-values 0.02, 0.04, 0.005, 0.004 and 0.004, respectively) while IL-33 was significantly associated with HDL-cholesterol and ILs 1α and NLRP3 (*p*-values 0.009, 0.004 and 0.007, respectively).

Lastly, bivariate correlations with NLRP3 (log-transformed) with ILs according to T2DM status and sex are presented in Appendix A, respectively.

## 4. Discussion

The present study assessed the circulating levels of NLRP3 and related ILs (1α, 1β, 33 and 37) in Saudi adults with PD and T2DM and observed that NLRP3 expression was significantly influenced by T2DM status and age, independent of sex, BMI and lipids, and was associated with ILs 18, 1α and 33. T2DM status also significantly altered 1L-1α levels after adjusting for the same covariates. When participants were stratified according to sex, the results showed higher levels of NLRP3 in the female PD group than their healthy counterparts, an observation not found in males. Further analysis using GLM, however, revealed no association between sex and the assessed NLRP3 and ILs. The presented results also showed a positive correlation between NLRP3 and both IL-33 and IL-37. Cumulatively, these results provide major insights into the role of NLRP3 in the progression of T2DM in this homogenous cohort. To the best of our knowledge, this is the first study to assess serum levels of NLRP3 along with several understudied ILs, such as IL-33 and -37, in a Saudi Arabian population with varying levels of glycemia.

It is worthy to note that in all participants, significantly higher levels of IL-37 were seen in the T2DM group as compared to PD and HC, even after adjusting for age and BMI. The latter findings were consistent with Li et al. in that IL-37 levels in senior Chinese participants with T2DM were overexpressed, and this was significantly and directly correlated with both insulin sensitivity and insulin resistance [20]. Furthermore, Hue et al. and Xie et al. found higher circulating levels of NLRP3 in PD participants compared to HC [21,22].

The significance of a cytosolic oligomer complex called the NLRP3 inflammasome has been recognized in the progression of insulin resistance [23] and in the development of T2DM [24,25]. Conclusive data have linked NLRP3 with the notorious harmful inflammatory complications of PD, which is the initial phase of T2DM development [23]. The NLRP3 inflammasome can be hyperactivated by a wide variety of DAMPs [26] (i.e., glucose, extracellular ATP, gout-linked etiologic agents such as calcium pyrophosphate dehydrate crystals and uric acid crystals, dietary saturated fatty acids, etc.) [27], resulting in the direct release of pro-inflammatory inflammasome-related ILs, e.g., 1β and 18 [26]. DAMPs arising from metabolic stress can also trigger the innate immune system and activate the NLRP3 inflammasome, stimulating the activation of stress and pro-inflammatory pathways in adipocytes [6]. Even though the activation of these pathways and the production of these proinflammatory ILs (IL-1β and IL-18) are natural protective measures, long-term chronic inflammation can exhaust the innate immune system and cause insulin resistance, hyperglycemia and lipidemia [28], linking NLRP3 inflammasomes between metabolic and immune responses associated with lipids and glucose metabolism [29].

After NLRP3 inflammasome activation, cells produce high levels of pro-inflammatory IL-1β and IL-18 [30]. IL-1β, in turn, up-regulates the secretion of insulin, amplifying the pro-inflammatory environment in macrophages and increasing the uptake and metabolism of glucose and the expression of insulin receptors, which are all detrimental to the metabolism. Increased uptake and metabolism of glucose by cells increases the production of reactive oxygen species [31] and hinders insulin signaling by decreasing covalent modification of insulin receptor substrate 1 (IRS-1) [32], thus hindering glucose sensitivity and insulin sensitivity and ultimately leading to the development of diabetes [30]. The active NLRP3 inflammasome also triggers insulin resistance via disturbance of AKT or the phosphatidylinositol 3-kinase/protein kinase B signaling pathway [6]. The NLRP3 inflammasome effector protein ‘caspase-1’; in addition to IL-1β and IL-18, also cleaves and activates the protein gasdermin D (GSDMD), allowing it to bind, oligomerize and form pores on the plasma membrane, feasibly compromising the integrity of the plasma membrane and resulting in a necrotic, pro-inflammatory type of cell death termed pyroptosis [9]. NLRP3 is a necessary protein in the pyroptosis pathway [33]. The transcription factor, nuclear factor kappa B (NF-κB), is involved via GSDMD in NLRP3 inflammasome-mediated cell death [34]. Pyroptosis releases all the intra-cellular components of the cell, including the inflammasome complex protein oligomers, into the circulation, amplifying the systematic inflammatory feature of T2DM [10]. Serum NLRP3 levels are used as possible inflammation biomarkers in different inflammatory diseases [33].

In the present study, circulating levels of NLRP3 were significantly higher in the female PD group after stratification, but after considering all variables in the GLM, it was found that sex as a factor in NLRP3 expression was not significant. This observation on the lack of sexual dimorphism may be limited by the sample size and should be interpreted with caution, as genetic variations and epigenetic modifications following exposure to different conditions such as T2DM susceptibility are consistently sexually dimorphic [35].

Wan et al. analyzed the gene expression of NLRP3 in peripheral blood mononuclear cells (PBMC) of T2DM individuals; their levels were considerably higher in T2DM, but were significantly reduced post-lifestyle changes and treatment therapy [25]. Similarly, Chen et al. found higher gene and protein levels of NLRP3 in PBMCs of patients with diabetic retinopathy [36]. Although the mentioned studies did not analyze the circulating levels of NLRP3, they still supported the results of our study, proposing a possible role of the NLRP3 inflammasome in T2DM. On the contrary, Dyer et al. analyzed NLRP3 gene expression in PBMCs in midlife T2DM and HC and found no significant differences [37]. A possible explanation for these contradicting results is that they only included T2DM patients with no further complications.

IL-37 is a newly discovered anti-inflammatory IL and a natural suppressor of the innate immune system [38]. It exerts protective effects on chronic inflammatory diseases [39] and may possibly be involved in the pathogenesis of many pathological conditions, including T2DM [40,41]. IL-37 is activated intracellularly by NLRP3 and caspase-1 cleavage. active IL-37 then forms a complex with other transcription factors (e.g., SMAD-3) and translocates to the nucleus, where it negatively regulates cytokine expression, thus decreasing inflammation [39]. IL-37 also exerts anti-inflammatory roles extracellularly: it binds to IL-1R8 and IL-18Rα and negatively regulates the MAPK and NF-κB pathways [38,39,41]. During inflammatory states, IL-37 decreases the production of numerous pro-inflammatory ILs while simultaneously increasing the production of other anti-inflammatory ILs [39]. Thus, IL-37 has a significant role in both human health and the progression of inflammatory diseases, and can be used as a potential biomarker.

As limited data on the circulating levels of IL-37 are available in the literature, our results are the first to investigate the signature levels of IL-37 in a group of both PD and T2DM and compare them to each other and to a HC group. Li et al. analyzed IL-37 mRNA and serum levels only in T2DM; as is consistent with our result, they found higher levels compared to HC [20]. Ye et al. observed higher levels of IL-37 in patients with hypertension [42]. However, there are some disagreements on the levels of IL-37 in the disease state, for example, Liu et al. found low levels of IL-37 in patients with cardiac disease and an inverse correlation between IL-37 and favorable prognosis [38]. However, Harms et al. observed no difference between type 1 diabetes mellitus (T1DM) patients and HCs [43]. High IL-37 levels maybe be a signifying marker in PD and T2DM patients, and IL-37 may be a novel biomarker and therapeutic target for preventing and treating DM as a whole. Further studies are needed to confirm our results and to clarify the role of IL-37 in health and disease.

IL-33 is a newfound cytokine, and is part of the IL-1 family. IL-33 is associated with the secondary innate immune response [44]. It has recently been found that NLRP3 exerts roles beyond the inflammasome: it functions as a transcription factor that can up-transcribe IL-33 by straight binding and activating its promotor [45]. Cell damage (necrosis) can induce the release of active IL-33 into the circulatory system. It is believed that IL-33 functions as an ‘alarming’ signal that is detected after cell death to warn the immune system of the presence of any kind of internal stress or tissue damage [46]. Despite the significant anti-inflammatory roles of IL-33, it seems that IL-33 is more complex and may have a dual role as both anti-inflammatory and pro-inflammatory, depending on the environment [44]. Many results have shown that IL-33 may have an unknown role in diabetes and cardiac complications of obesity because the precise effect of IL-33 in the progress of many metabolic diseases, including diabetes, seems to be unpredictable [44].

Our results showed a positive correlation between circulating levels of NLRP3 and IL-33 in all participants (Figure 2A) and in all three (HC, PD, T2DM) groups (Appendix A). These correlations confirm that NLRP3 independently exerts a role as a transcription factor [45]. NLRP3 forms a complex with interferon regulatory factor 4 protein and directly binds to the IL-33 promotor region, up-regulating its expression [45]. IL-33 in human adipocytes was also observed to up-regulated in T2DM patients’ subcutaneous adipose tissue (SAT) [47]. Other studies found that IL-33 was upregulated in T2DM patients with chronic hepatitis B [48]. To the best of our knowledge, no studies have analyzed the circulating IL-33 in individuals with PD or T2DM, and ours is the first to investigate this.

IL-1α is both a DAMP and an alarming cytokine released from pyroptotic cells as an activator of sterile inflammation [49]. Involved in various inflammatory diseases in humans, IL-1A remains a relatively obscure cytokine that certainly warrants additional research [50], signifying the importance of the results of the current cross-sectional study. The study found no significant differences in circulatory levels of IL-1α between all three groups (HC, PD, T2DM). However, Salti et al.’s results revealed that high glucose levels induce IL-1α production, but this occurs in diabetic nephropathy patients [49]. The results of this study also showed that circulatory NLRP3 levels significantly predict IL-1α levels and vice versa; IL-1α significantly predicts NLRP3 levels (Table 2). In support of these results, Dagvador et al. found that IL-1α-deficient macrophages had decreased caspase-1 activity and, thus, reduced IL-1β release. This elucidates IL-1α’s role in activating the NLRP3 inflammasome complex [51], and in predicting NLRP3 levels, as found in this cross-sectional study.

The authors acknowledge some limitations. First, the cross-sectional study design only took into consideration one time point, and, therefore, causality cannot be inferred. Second, although it was mentioned in the discussion, we did not analyze the gene expression of the NLRP3 inflammasome, other related genes such as PGC1α and AKT nor other markers related to pyroptosis, which also play a significant role in T2DM complications [52,53]. Third, we did not assess the activity of the NLRP3 inflammasome to rule out the effect of other pathways on related ILs. Lastly, although the sample size was adequate, multiple stratifications and adjustments may have limited the study to consistently elicit differences in males and females. Nevertheless, the current cross-sectional study is the first of its kind to elicit differences and associations in NLRP3 with several emerging ILs of the inflammasome complex in a homogenous ethnic population with varying levels of dysglycemia.

## 5. Conclusions

In conclusion, T2DM status significantly affects levels of NLRP3 and other ILs to varying degrees, independent of sex, BMI and other known clinical factors. These results suggest that glycemic status plays a major role in influencing the activity of the inflammasome complex. Prospective studies are needed as to whether improving glycemic status favorably alters NLRP3 and other emerging regulators of the innate immune system.

## Figures and Tables

**Figure 1 biomedicines-11-01315-f001:**
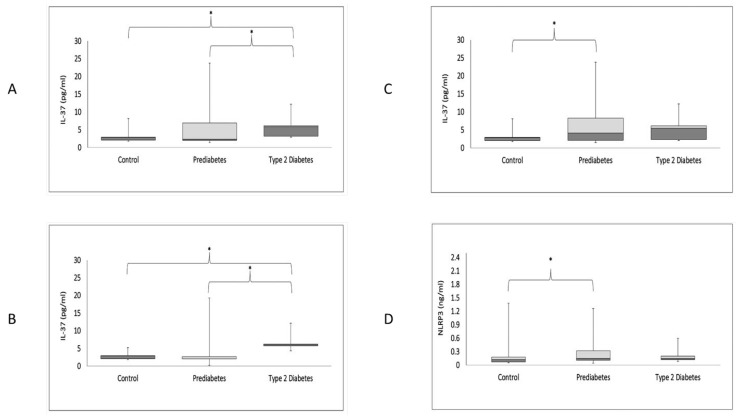
Boxplots showing circulating levels of IL-37 in all participants (**A**), males (**B**) and females (**C**); and NLRP3 in females (**D**). The light gray and dark gray boxes show the third and first quartiles, respectively, and the meeting point signifies the median. * denotes *p* < 0.05.

**Figure 2 biomedicines-11-01315-f002:**
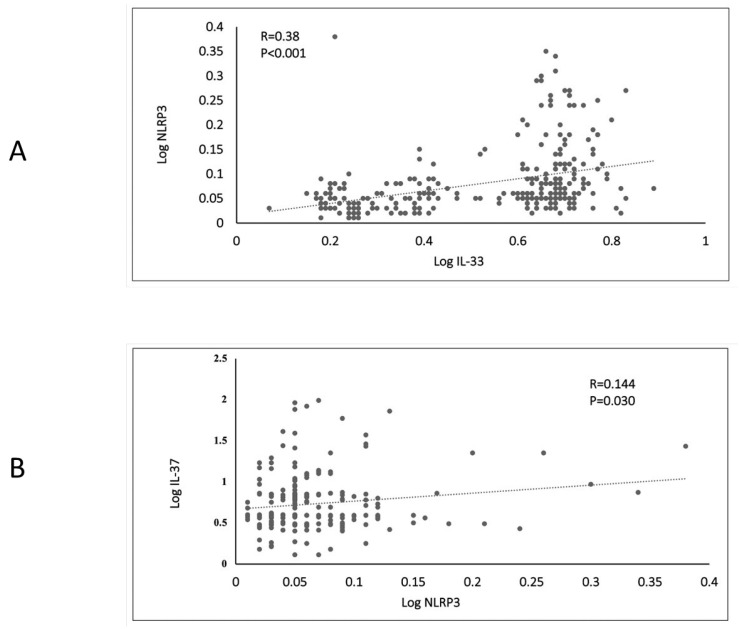
Scatterplots representing the bivariate correlations of NLRP3 with IL-33 (**A**) and IL-37 (**B**) in all participants. Variables were log-transformed before the correlation analysis was performed. A *p* < 0.05 was considered significant.

**Table 1 biomedicines-11-01315-t001:** General characteristics of participants, according to T2DM status.

Parameters	HC	PD	T2DM	*p*-Value	*p*-Value (Age and BMI Adjusted)
N (M/F)	198 (87/111)	88 (38/50)	121 (26/95)		
Age (year)	39.3 ± 8.9	42.3 ± 8.8 ^A^	44.0 ± 8.7 ^A^	<0.001	
BMI (kg/m^2^)	28.7 ± 5.3	32.7 ± 5.9 ^A^	32.6 ± 7.2 ^A^	<0.001	
WHR	0.85 ± 0.09	0.87 ± 0.08	0.90 ± 0.09 ^A^	0.001	
Systolic blood pressure (mmHg)	119.2 ± 12.0	122.4 ± 12.7	127.5 ± 16.2 ^AB^	<0.001	0.001
Diastolic blood pressure (mmHg)	74.8 ± 9.9	76.6 ± 9.9	78.5 ± 10.3	0.009	0.16
Fasting glucose (mmol/L)	4.81 ± 0.50	5.98 ± 0.30 ^A^	8.62 ± 2.50 ^AB^	<0.001	<0.001
Hb1AC (%)	5.0 ± 0.7	5.6 ± 0.5 ^A^	7.2 ± 1.4 ^AB^	<0.001	<0.001
Insulin (uU/mL)	12.7 (6.3–18.9)	15.9 (15.6–16.7)	19.4 (15.9–19.8) ^AB^	<0.001	<0.001
Total cholesterol (mmol/L)	5.2 ± 0.9	5.1 ± 0.9	5.3 ± 1.3	0.268	0.24
HDL cholesterol (mmol/L)	1.3 ± 0.4	1.1 ± 0.3 ^A^	1.2 ± 0.4 ^A^	<0.001	<0.001
Triglycerides (mmol/L)	1.2 (0.9–1.7)	1.7 (1.2–2.3) ^A^	1.7 (1.3–2.2) ^A^	<0.001	<0.001

Note: Data are presented as mean ± SD and median (25th–75th) percentiles for normal and non-normal variables, respectively. Superscripts A and B represent *p*-values found to be significant for HC and PD by post hoc analysis at *p* < 0.05.

**Table 2 biomedicines-11-01315-t002:** Significant predictors of NLRP3 and select ILs.

	NLRP3	IL-1α	IL-33
Adjusted R^2^ (explained variation)	0.46	0.43	0.11
Independent predictors	IL-1α (β = 0.18), Triglycerides (β = 0.09)	NLRP3 (β = 2.45), Triglycerides (β = 0.09)	Diastolic BP (β = −0.01)

Note: Data are presented as adjusted r-squared obtained from stepwise regression analysis. *p* < 0.05 was considered significant.

**Table 3 biomedicines-11-01315-t003:** General linear model (GLM) using T2DM status as the dependent variable and gender, age, anthropometrics, glucose, lipids, NLRP3 and ILs as independent variables.

Dependent Variable	NLRP3		IL-1α		IL-33	
Independent Variables	β ± S.E.	*p*-Value	β ± S.E.	*p*-Value	β ± S.E.	*p*-Value
Group status						
HC	1.0		1.0		1.0	
PD	−0.014 ± 0.01	0.33	−0.01 ± 0.04	0.84	0.140 ± 0.075	0.06
T2DM	−0.037 ± 0.02	0.028	0.08 ± 0.03	0.018	0.057 ± 0.065	0.38
Gender						
Female	1.0		1.0		1.0	
Male	0.008 ± 0.01	0.51	−0.047 ± 0.03	0.07	−0.062 ± 0.051	0.22
Age	−0.001 ± 0.0006	0.037	0.002 ± 0.001	0.20	0.003 ± 0.002	0.25
BMI	0.0001 ± 0.001	0.59	−0.001 ± 0.002	0.45	−0.004 ± 0.002	0.11
WHR	−0.032 ± 0.06	0.59	0.122 ± 0.14	0.38	−0.008 ± 0.27	0.98
Systolic BP	0.0001 ± 0.0003	0.65	0.00001 ± 0.0007	0.46	0.00001 ± 0.001	0.83
Diastolic BP	0.0002 ± 0.0005	0.84	0.00001 ± 0.001	0.71	−0.002 ± 0.002	0.44
Glucose	0.003 ± 0.004	0.40	−0.007 ± 0.008	0.40	−0.008 ± 0.016	0.64
HbA1C (%)	0.008 ± 0.006	0.17	0.004 ± 0.013	0.74	−0.045 ± 0.025	0.07
Cholesterol	−0.005 ± 0.004	0.24	0.003 ± 0.010	0.81	−0.007 ± 0.021	0.72
HDL cholesterol	−0.002 ± 0.02	0.86	0.011 ±0.032	0.75	−0.163 ± 0.062	0.009
Triglycerides	0.071 ± 0.04	0.07	−0.026 ± 0.093	0.78	−0.016 ± 0.179	0.93
IL-18	−0.022 ± 0.007	0.005	0.015 ± 0.018	0.41	−0.062 ± 0.034	0.08
IL-1α	0.132 ± 0.046	0.004	----	---	−0.577 ± 0.202	0.004
IL-1β	0.0001 ± 0.024	0.99	0.098 ± 0.056	0.08	0.009 ± 0.110	0.93
IL-33	0.064 ± 0.023	0.007	−0.156 ± 0.054	0.004	---	---
IL-37	−0.002 ± 0.016	0.92	0.002 ± 0.037	0.96	−0.068 ± 0.072	0.35
NLRP3	----	---	0.702 ± 0.243	0.004	1.269 ± 0.470	0.007

Note: Data are presented as β ± standard error (S.E.). *p* < 0.05 was considered significant.

## Data Availability

Data can be obtained upon request from the corresponding author.

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
