# Peer review of "Differences and Associations of NLRP3 Inflammasome Levels with Interleukins 1α, 1β, 33 and 37 in Adults with Prediabetes and Type 2 Diabetes Mellitus"

_biomedicines, 2023, doi:10.3390/biomedicines11051315_

Round 1
Reviewer 1 Report
Comments to the author:
The Manuscript " Differences in NLRP3 Inflammasome Levels Reveal Sexual Dimorphism in Adults with Prediabetes and Type 2 Diabetes Mellitus”. The results describe significant positive correlations between NLRP3 and both 29 IL-33 and IL-37. A unique sex-specific signature of NLRP3 and IL-37 in PD and T2DM individuals 30 was established for the first time
Major revisions required
Comments:
1- The protein expression of NRLP3 should be measured in serum by protein as well as IL by Elisa
2-Please explain the mechanism which correlates the expression of NRLP3 with the ILs
3- Please describe the expression of other genes like PGC 1alpha, AKT and IR which plays very important role
English may be improved
Author Response
Major revisions required
- The protein expression of NRLP3 should be measured in serum by protein as well as IL by Elisa.
Response: Great point, to clarify we already analyzed serum NLRP3 protein levels using a commercially available ELISA kit catalog number CSB-E15885h.
- Please explain the mechanism which correlates the expression of NLRP3 with the ILs
Response: Thank you for your valuable input. We have addressed this comment and tracked it in the attached article.
- Please describe the expression of other genes like PGC 1alpha, AKT and IR which plays very important role.
Response: Thank you for your valuable input. We have addressed this comment and described the expression of other genes including AKT, NFKB, IRS-1and tracked it in the attached article.
Reviewer 2 Report
I have studied carefully the manuscript entitled "Differences in NLRP3 Inflammasome Levels Reveal Sexual Dimorphism in Adults with Prediabetes and Type 2 Diabetes Mellitus" by Alfadul H. et al.
The manuscript deals with an interesting topic though underrecognized topic; thus, its potential publication could well benefit not only the specialists, namely the diabetologists and the rheumatologists/clinical immunologists, but also a broader readership.
The manuscript is generally well prepared; the language and the syntax are proper; the tables/figures are clear and informative. The references are all relevant and up-to-date.
However, before considering publication, the authors are wellcome to discuss the points listed below.
Major issues:
1) The statistical approach used could be well ameliorated, given that the aim of the study is to reveal any potential sex-based discrepancy. More specifically, a generalized Linear Model could further clarify both within and between independent parameters variability, thus enlightening the role of sex in the diabetic and pre-diabetic phenotype. In such a model, healthy ontrols (HC), prediabetes (PD), and type 2 diabetes (T2D) could be incorporated as dependent variables, while sex, age, body mass index (BMI), HbA1c, insulin as well as any other parameter that is believed to be either explanatory (NLRP3, IL-1A, IL-1β, IL-33, and IL-37) or confounder could be treated as independent ones.
Minor issues
1) The authors are wellcome to refer to sexual dimorphism in relation with genetic and epigenetic mechanisms in both the "Introduction" and the "Discussion" section. Some closely related references could be added, too (e.g Tramunt B. et al, Sex differences in metabolic regulation and diabetes susceptibility. Diabetologia 2020;63:453-461. PMID: 31754750).
2) The role of NLRP3 inflammasome could be further discussed in relation with insulin resistance (please see: Jorquera G. et al, NLRP3 Inflammasome: Potential Role in Obesity Related Low-Grade Inflammation and Insulin Resistance in Skeletal Muscle. Int J Mol Sci. 2021;22:3254. PMID: 33806797).
3) A clinical model would be futher supportive of any discussion and/or conclusions. Are there any clinical data concerning autoinflammatory diseases involving NLRP3 molecular pathophysiology, such as Familial Mediterranean Fever?
The language and the syntax are proper; only minor editing is required.
Author Response
Major issues:
- The statistical approach used could be well ameliorated, given that the aim of the study is to reveal any potential sex-based discrepancy. More specifically, a generalized Linear Model could further clarify both within and between independent parameters variability, thus enlightening the role of sex in the diabetic and pre-diabetic phenotype. In such a model, healthy controls (HC), prediabetes (PD), and type 2 diabetes (T2D) could be incorporated as dependent variables, while sex, age, body mass index (BMI), HbA1c, insulin as well as any other parameter that is believed to be either explanatory (NLRP3, IL-1A, IL-1β, IL-33, and IL-37) or confounder could be treated as independent ones.
Response: Thanks to reviewer valuable comments we have run the Generalized Linear Model according to instruction above and available as supplementary table (S3).
Minor issues
- The authors are welcome to refer to sexual dimorphism in relation with genetic and epigenetic mechanisms in both the "Introduction" and the "Discussion" section. Some closely related references could be added, too (e.g Tramunt B. et al, Sex differences in metabolic regulation and diabetes susceptibility. Diabetologia 2020;63:453-461. PMID: 31754750).
Response: Thank you for your valuable input. We have addressed this comment and tracked it in the attached article (lines 289-291).
- The role of NLRP3 inflammasome could be further discussed in relation with insulin resistance (please see: Jorquera G. et al, NLRP3 Inflammasome: Potential Role in Obesity Related Low-Grade Inflammation and Insulin Resistance in Skeletal Muscle. Int J Mol Sci. 2021;22:3254. PMID: 33806797).
Response: Thank you for your valuable input. We have addressed this comment in the discussion section and tracked it in the attached article (lines 252-266).
- A clinical model would be further supportive of any discussion and/or conclusions. Are there any clinical data concerning autoinflammatory diseases involving NLRP3 molecular pathophysiology, such as Familial Mediterranean Fever?
Response: We agree with the suggestion, unfortunately, we don’t have such supportive data. The exclusion criteria that were taken into consideration included pre-existing inflammatory systemic disease.
Round 2
Reviewer 1 Report
It can be accepted in current form
It can be accepted in current form
Author Response
We sincerely thank the reviewer for appreciating our work.
Reviewer 2 Report
I have carefully studied the revised version of the manuscript entitled "Differences in NLRP3 Inflammasome Levels Reveal Sexual Dimorphism in Adults with Prediabetes and Type 2 Diabetes Mellitus" by Alfadul et al.
The authors have responded to all queries raised and performed all necessary alterations/corrections requested.
However, the use of the proposed General Linear Model (GLM) failed to reveal a clear-cut sexual dimorphism in adults with Prediabetes and Type 2 Diabetes Mellitus as far as NLRP3 Inflammasome Levels are concerned. In detail, by examining Supplementary Table 3, where the results of the GLM are provided, we can conclude that sex, being treated as a potential explanatory variable, affects neither NLRP3 (P=0.51), nor IL-1α (P=0.07), nor IL-33 (P=0.22).
As a matter of fact, the authors are requested to reconsider revising their Title, Abstract, Results, Discussion, and Conclusions in such a way that "sexual dimorphism" will not be the key finding, thus avoiding conveying a potentially misleading message.
Only minor language editing is required.
Author Response
I have carefully studied the revised version of the manuscript entitled "Differences in NLRP3 Inflammasome Levels Reveal Sexual Dimorphism in Adults with Prediabetes and Type 2 Diabetes Mellitus" by Alfadul et al. The authors have responded to all queries raised and performed all necessary alterations/corrections requested. However, the use of the proposed General Linear Model (GLM) failed to reveal a clear-cut sexual dimorphism in adults with Prediabetes and Type 2 Diabetes Mellitus as far as NLRP3 Inflammasome Levels are concerned. In detail, by examining Supplementary Table 3, where the results of the GLM are provided, we can conclude that sex, being treated as a potential explanatory variable, affects neither NLRP3 (P=0.51), nor IL-1α (P=0.07), nor IL-33 (P=0.22). As a matter of fact, the authors are requested to reconsider revising their Title, Abstract, Results, Discussion, and Conclusions in such a way that "sexual dimorphism" will not be the key finding, thus avoiding conveying a potentially misleading message.
Response: We sincerely appreciate the reviewer's concern and the legitimate point raised. We have substantially amended the necessary segments of the paper to eliminate the sexual dimorphism parts which were not supported by the GLM analysis. We have also moved table S3 to the main manuscript as table 3. The revised version now conveys a clear message fully supported by the findings. We hope these changes will now merit favor and again we are truly thankful for this major point.
Round 3
Reviewer 2 Report
I have studied the second revision of the manuscript entitled "Differences and Associations of NLRP3 Inflammasome Levels with Interleukins 1α, 1β, 33 and 37 in Adults with Prediabetes and Type 2 Diabetes Mellitus" by Alfadul et al.
The authors have thoroughly revised the manuscript according to the major query raised.